# Electrochemical Signal Amplification Strategies and Their Use in Olfactory and Taste Evaluation

**DOI:** 10.3390/bios12080566

**Published:** 2022-07-26

**Authors:** Xinqian Wang, Dingqiang Lu, Yuan Liu, Wenli Wang, Ruijuan Ren, Ming Li, Danyang Liu, Yujiao Liu, Yixuan Liu, Guangchang Pang

**Affiliations:** 1Tianjin Key Laboratory of Food Biotechnology, College of Biotechnology & Food Science, Tianjin University of Commerce, Tianjin 300134, China; wangxinqian@stu.tjcu.edu.cn (X.W.); liming@tjcu.edu.cn (M.L.); 18731806261@163.com (D.L.); m15568219515@163.com (Y.L.); liuyixuan@stu.tjcu.edu.cn (Y.L.); 2Department of Food Science & Technology, School of Agriculture & Biology, Shanghai Jiao Tong University, Shanghai 200240, China; y_liu@sjtu.edu.cn (Y.L.); wenli-wang@sjtu.edu.cn (W.W.); 3Tianjin Institute for Food Safety Inspection Technology, Tianjin 300308, China; renruijuan2013@126.com

**Keywords:** electrochemical biosensors, olfactory and taste evaluation, signal amplification strategies, nanomaterials, enzymes, nucleic acid amplification techniques

## Abstract

Biosensors are powerful analytical tools used to identify and detect target molecules. Electrochemical biosensors, which combine biosensing with electrochemical analysis techniques, are efficient analytical instruments that translate concentration signals into electrical signals, enabling the quantitative and qualitative analysis of target molecules. Electrochemical biosensors have been widely used in various fields of detection and analysis due to their high sensitivity, superior selectivity, quick reaction time, and inexpensive cost. However, the signal changes caused by interactions between a biological probe and a target molecule are very weak and difficult to capture directly by using detection instruments. Therefore, various signal amplification strategies have been proposed and developed to increase the accuracy and sensitivity of detection systems. This review serves as a reference for biosensor and detector research, as it introduces the research progress of electrochemical signal amplification strategies in olfactory and taste evaluation. It also discusses the latest signal amplification strategies currently being employed in electrochemical biosensors for nanomaterial development, enzyme labeling, and nucleic acid amplification techniques, and highlights the most recent work in using cell tissues as biosensitive elements.

## 1. Electrochemical Biosensors

In the 1960s, Leland C. Clark Jr, an American scholar in electroanalytical chemistry, suggested that that the determination of biochemicals could be found using a method as convenient as pH electrodes, which led to the introduction of enzyme electrodes, the first biosensors [1,2,3]. For half a century, biosensing has developed into a classic converging technology with the incorporated principles and technologies of multiple disciplines such as life sciences, chemistry, physics, information, and materials [4]. In the 1970s to 1980s, various biomolecules and biomaterials were used as the molecular recognition elements for biosensors, enabling the rapid detection of a variety of biochemical and immunological substances [4]. In addition, various physical and chemical transduction principles were adopted, driving the formation of the biosensing field. In the second wave of development, second-generation enzyme electrodes were commercially successful [4], surface plasmon resonance (SPR) biosensors were widely used for biomolecular interaction studies [5], while DNA microarrays enabled high-throughput analysis of gene expression [4]. Since the 21st century, the introduction of nanotechnology has endowed biosensing with many new properties such as high sensitivity, a multiparameter nature, and microenvironmental applications [6]. Biosensors are powerful pieces of analytical equipment used to identify and detect target molecules, and are usually composed of a biosensing material and a physicochemical sensor [7]. Biosensors are generally used as detectors, and utilize a bioactive substance as a biofunctional sensitive element fixed to a signal transducer, which transmits a signal that is then converted to corresponding optical, thermal, and electrical signals with good sensitivity, selectivity, and specificity when a specific target is added. However, a bottleneck in the application of the biological receptor elements is the maintenance of their vitality, stability and shelf-life upon bonding with the electronic elements [8].

Among the known types of biosensors, electrochemical biosensors are efficient analytical tools that combine biosensing and electrochemical analysis techniques [9] and are generally built in three-electrode electrochemical cells that consist of a working electrode, a counter electrode, and a standard electrode with a stable and fixed potential [10]. Analytical methods for electrochemical biosensors are usually based on the electron transfer process between an electrode surface and an electroactive material in an electrolyte [7]. Electrochemical biosensors use a fixed electrode as the base electrode and fixed bioactive molecules on their surfaces, capturing target molecules onto the electrode surface through specific recognition between biomolecules, where the base electrode converts the concentration signal into measurable electrical signals such as current, potential, and resistance. This enables the quantitative and qualitative analysis of a target. The basic principle of electrochemical biosensors is shown in Figure 1. Four signal conversion types exist for electrochemical biosensors: current, potential, impedance, and ion charge (field effect). Among the current-based electrochemical biosensors, commonly used detection methods include cyclic voltammetry (CV) [11], square wave voltammetry (SWV) [12], differential pulse voltammetry (DPV) [11], and electrochemical impedance spectroscopy (EIS) [12]. Electrochemical biosensors have been widely studied in simple or complex detection environments and in a variety of fields due to their high selectivity for the molecules they can identify, as well as their high sensitivity [13], fast response time, miniaturized and portable properties [14], compatibility with impurity matrices [15,16], simplicity of operation, and low cost [17,18].

Although electrochemical biosensors are already highly sensitive, their sensitivity must be further improved for the detection of certain molecules at low concentrations or molecules that are difficult to isolate from biological samples [19,20,21,22]. In recent years, detection methods for specific interactions between biological recognition elements, such as antibodies, nucleotides, enzymes, and target analytes have been proposed and developed to improve the sensitivity and selectivity of detection systems [23]. In addition, signal amplification technology is often used as a critical technology in biosensor manufacturing because it plays a crucial role in improving the sensitivity, selectivity and stability of biosensors. Several signal amplification strategies, such as the use of nanomaterials with unique physicochemical properties, as well as the use of enzymatic labeling and nucleic acid amplification techniques, have become widespread. This review will present several aspects of signal amplification strategies commonly used in electrochemical biosensors (Figure 2), as well as present recent results regarding their use in olfactory and taste determination.

## 2. Advances in Electrochemical Signal Amplification Strategies for Olfactory and Taste Measurements

The detection of gases, including malodorous molecules and volatile organic compounds (VOCs), has attracted great interest in recent years and there has been a growing demand for it in various fields. Volatile organic compounds (VOCs) are a large class of low molecular weight (<300 Da) carbon-containing compounds. These small volatile molecules have a wide range of sources, both natural (plants, animals, bacteria, etc.) and anthropogenic (fossil fuels, automobile exhaust, etc.) [24]. Studies have shown that most VOCs have adverse effects on human health, causing symptoms such as headaches, and nose, eye and throat irritation [25]. They are also considered chemical messengers, and studies have identified different gases associated with different diseases. In addition, VOC and odor analysis can be used for quality assessment in the food, beverage, and flavor industries. Therefore, it is crucial to monitor the nature and concentration of these compounds in indoor or outdoor environments [24].

### 2.1. Classical Analytical Techniques for Olfactory and Taste Detection

Olfactory and taste sensation are widespread in nature, and they play a major role in the survival and reproduction of natural organisms [26,27]. Olfactory and taste receptors mainly consist of cellular, tissue, or biological sensing receptors for various signals around the body, especially for food and its nutrients. Studies have shown that olfactory perception and taste are dependent on the sensing effect of G-protein-coupled receptors (GPCRs), making them the most important targets for drug screening. GPCRs are a superfamily of thousands of members that plays an extremely important role as nutrient sensor receptors in the metabolism of substances, capacity metabolism, and signal communication in the body or cells [28]. Methods commonly used for olfactory and taste detection include gas chromatography (GC), gas chromatography-mass spectrometry (GC-MS), electronic nose (EN), electronic tongue (ET) [29], near-infrared spectrum (NIR), and other biosensors based on olfactory receptors (OR) or taste receptors (TR) [30,31]. In addition, natural elements such as odor-binding protein (OBP) or its analogs, such as peptides, are often used in the construction of olfactory electrochemical biosensors [8,24,32,33].

GC is a separation and analysis method using gas as a mobile phase, which has the advantages of high separation efficiency, fast analysis, high sensitivity and good selectivity, etc. [34]. It has been widely used in various fields and plays an important role in various aspects of modern society. GC consists of five systems: the gas circuit system, the sample injection system, the separation system, the temperature control system and the detection and recording system, of which the separation system and the detection and recording system are the core. With an inert gas as the mobile phase, GC takes advantage of the fact that the partition coefficient of components in a sample varies with the gas and stationary phases [34]. As the sample is carried into the column by the carrier gas, the components undergo repeated alternating distribution between the two phases. The components in the stationary phase have different absorption capacities and therefore the analytes pass through the column at different rates. After a certain column length, the components of the sample are separated from each other and enter the detector. The ion current signal generated by each component is amplified to produce a peak for each component. In this way, the purpose of separation and detection is achieved. Gas chromatography-mass spectrometry (GC-MS), a highly sensitive and accurate analytical technique, allows the separation, identification, and quantification of different VOCs in a mixture [34]. NIR is an electromagnetic spectrum between visible light (VIS) and mid-infrared (MIR). NIR is a method that uses chemical bonds containing hydrogen groups to stretch and relax frequencies, resulting in vibrational and combined frequencies. Fourier transform infrared spectroscopy (FTIR) requires a spectrometer and scanning means to analyze gases. FTIR can measure and analyze the concentration of toxic gases in a wide range of infrared regions [34].

Although GC and GC-MS have good odor detection capabilities, they are not olfactory sensors. In addition, they are bulky and expensive and require highly time-consuming laboratory operations [34]. FTIR is highly sensitive and enables simultaneous analytical measurements of multiple gases, but its gas measurement and analysis can only be performed in the laboratory, thus making it impossible to achieve online real-time gas detection using FTIR [34]. Such a background has prompted many researchers to work on developing alternative techniques to overcome the various drawbacks mentioned above. Therefore, there is a need for an affordable, reliable, portable and sensitive device that can rapidly analyze gases, including VOCs [24].

### 2.2. Olfactory and Taste Detection Based on Biosensor Technology

Electronic noses and electronic tongues can detect odors and taste by using chemically sensitive materials and are based on chemical interactions [35,36,37]. They are fast, simple, and portable detection tools. However, they are unable to distinguish between chemicals with similar structures [37]. Linda B. Buck and Richard Axel have conducted numerous research efforts on biological olfaction, which has shown that in order to distinguish between various odors, biological noses exploit the cross-reactivity of olfactory receptors (OR), prompting each receptor to interact with different odor molecules [38].Thus, as with barcodes, odors are encoded by combinations of olfactory receptors, prompting the nose to have a wide detection range [24]. In addition, Linda B. Buck and Richard Axel demonstrated that ORs belong to the large family of G-protein-coupled receptors (GPCRs) [24]. In bioelectronic noses (B-EN) and bioelectronic tongues (B-ET), more selective odor detection can be achieved by using specific receptors, and transistor-based nanomaterials can be used to amplify sensory signals, such as carbon nanotubes [39], conductive polymers [40], and GR [41]. Bioelectronic noses-based nanomaterials are small, highly portable, and can be used for field odor analysis by combining them with portable current measurement devices [42]. In recent years, bioelectronic devices that use human sensory receptors as molecular recognition elements have been developed and have been commonly used to characterize food quality and safety. In addition, multi-channel bioelectronic noses have been developed that consist of arrays of olfactory receptors capable of individually analyzing various odor information [42,43]. However, most current EN systems use chemical layers as sensing elements and, therefore, have the disadvantage of limited diversity of sensor coatings and poor selectivity [24]. In addition, EN, B-EN, ET, and B-ET instruments not only have the disadvantage of a lack of sensor stability, but also the difficulty of having identical sensing characteristics of the instrument in different production batches. Recent research trends suggest that natural elements, such as ORs, OBP and peptides, can also be used as sensitive materials in biosensors to improve odor sensing performance [44,45].

Since the discovery of the vertebrate olfactory receptor (ORs) gene family by Buck and Axel, much progress has been made in the study of the molecular mechanisms of olfaction and signal transduction pathways [31,38]. The process of biological olfactory is the selective recognition of odors by ORs, triggering the intracellular signal transduction pathways that lead to the depolarization of the OSN, and ultimately the transmission of information to the brain for processing via neuronal axonal connections. Olfactory biosensors use similar signal transduction mechanisms to recognize different odors and convert odor chemical signals into readable signals, such as electrical, and optical signals [31]. Technologies such as microelectrodes, light addressable potential sensors (LAPS), field effect transistors (FETs), and electrochemical impedance spectroscopy (EIS) have been commonly used in OR-based electrochemical biosensors for olfactory signal conversion. FETs have inherent signal amplification, making them particularly suitable for the detection of weak signals in OR-based biosensors [31].

In biosensors, the characteristics and properties of the sensing material should be maintained by adopting an appropriate immobilization strategy, depending on the biometric parts. Methods that couple biological elements to appropriate transduction systems can often be used to obtain measurable and detectable signals [8]. However, combining ORs with signal transducers is a major challenge as ORIS membranes are bound to volatile organic compounds (VOCs), making it difficult to obtain configurations where ORs can bind to VOCs under membrane-free conditions [8]. To facilitate the incorporation of ORs into sensors, six materials have been typically utilized for binding, including cells and tissues, nanovesicles, nanodiscs, artificial lipid bilayers, odorant binding proteins, and biomimetic materials [8]. When embedding ORs using nanovesicles, the cell membrane containing an OR can be constructed as a nanoscale phospholipid bilayer structure to maintain the natural environment of the OR [46,47]. In addition, when the OR is integrated into a sensor using nanodiscs, these nanodiscs can provide a stable environment for ORs. However, this will separate the OR from the downstream proteins that promote the olfactory process [47]. Artificial lipid bilayers can simulate cell membranes, maintaining intact membrane proteins, such as ORs, and keeping their function intact. Biomimetic materials allow for the immobilization of synthetic peptides based on ORs and OBP rather than entire proteins, as they do not require tertiary structures or lipid membranes, thus improving stability and repeatability. Artificial receptors, such as molecularly imprinted polymers (MIP), can also be used as sensing elements due to their high stability [8].

Compared to traditional odor analysis techniques, electronic noses, electronic tongues, and olfactory and gustatory electrochemical biosensors are fast, convenient, and economical, and are widely used in food, medicine, agriculture, and environmental monitoring [34]. In addition, gas sensing plays an important role in security applications (detection of drugs, explosives, etc.), environmental monitoring, and other applications under development, such as augmented or virtual reality [48]. Nevertheless, research on biosensors for olfaction and taste determination is still in the early experimental stages and further research is still needed as commercial OR biosensors are not yet available in the market due to the fragility of biosensing elements and the lack of portable signal transduction systems.

### 2.3. Taste Electrochemical Sensors Based on a Cellular Signal Cascade Amplification System

Currently, most studies that are focused on taste receptor sensors are based on changes in ion channels such as Ga^2+^ inward flow resulting from receptor–ligand interactions in living cells, which are dependent on a variety of complex factors such as cell type, physiological activity, environment, and intercellular interactions [34]. Although molecular interaction instruments based on SPR technology that can be used to detect non-standard receptor–ligand interactions are on the market, they are costly, technically complex, and struggle to achieve high throughput, making conventional taste detection difficult [34]. In current research, the methods for quantifying taste sensation have usually been based on three mechanisms, namely: labeling based on ion channels or cellular active components in living cells, non-standard-SPR methods, where the binding and dissociation properties of receptor–ligands are measured, and electronic tongues. In modern biotechnology, multiple molecular signal transduction components can be co-expressed in heterogeneous cell systems, thereby converting chemical signals into electrical signals [31].

Lee et al. [49] designed a miniature planar electrode to record the general membrane potential changes of a heterogeneous olfactory system based on the co-expression of ORI7 and taste cyclic nucleotide gate (CNG) channels in HEK-293 cells. An olfactory biosensor based on the multipoint detection of the electrical activity of olfactory cells and tissues combined signal processing methods with olfactory decoding theory and showed excellent potential for the simultaneous detection of multiple odors in complex environments with high sensitivity and selectivity [31]. Xu et al. [50] constructed a novel hGPR120 fatty acid receptor sensor based on the self-assembly of the hGPR120 receptor onto the surface of bilayer-modified gold nanoparticles and bovine taste buds, which successfully detected the G protein signals generated by the interaction of this sensor with 14 different natural fatty acids. Pang et al. [34] self-assembled the T1R1 umami receptor protein expressed in vitro by rats onto nanogold, and constructed an electrochemical biosensor based on signal amplification using horseradish peroxidase for the quantitative determination of glutamate monosodium salt concentrations.

An electrochemical taste sensor based on cell signal amplification has the advantages of high sensitivity, strong specificity, quantification, simple operation, low price, and good repeatability, and this technology can provide a good platform for the study of GPCRs and their interaction patterns with ligands and biological functions, which can then be used for taste determination in humans and animals [34].

### 2.4. Olfactory Electrochemical Sensors Based on a Cellular Signal Cascade Amplification System

Compared to other detection methods, OR electrochemical sensors based on the signal cascade amplification systems of the cell itself should be more sensitive by several orders of magnitude for detecting their respective ligands [51]. Studies have shown that ORCO expression will lead to an increase in ligand sensitivity and a decrease in the lower limit of detection [52]. The use of OR-expressing cells and tissues as biosensing elements is based on the generation of signal cascades of ions transferred from the outside to the inside of cells caused by the combination of OR odor [46]. In addition, OR electrochemical sensors can be coupled with enzymatic and nanomaterial electrochemical signal amplification methods. Given the natural diversity of the olfactory system and the compounds that can bind to gas molecules, a wider range of biosensors can be constructed for different application aspects [8]. Lu et al. [31] prepared a sandwich electrochemical olfactory sensor to detect sex differences in male and female rats, using the vomeronasal organ tissue of rats as a reference. The results showed that the vomeronasal organ sensors of male and female rats had different dynamic curves for their respective urine and were able to distinguish between their own urine and the urine from other rats. Lu et al. [53] simulated intracellular receptor signal processes based on an electrochemical signal amplification system of gold nanoparticles (AuNPs) and HRP. Using gold nanoparticles self-assembled twice and the subsequent adsorption of Bombyx olfactory receptor 1 (BmOR1), a sex pheromone binding protein, an electrochemical upper nanogold membrane receptor sensor was constructed. Kang et al. [54] constructed an H_2_O_2_ electrochemical biosensor based on the nanogold adsorption of immobilized horseradish peroxidase HRP with thionine-chitosan as a bridging agent. On this basis, a bilayer nanogold-modified *Bacillus cereus* immunosensor was prepared based on the nanogold adsorption of HRP for electrochemical signal amplification using *Bacillus cereus* monoclonal antibody as the biomolecular recognition element and chitosan as the bridging agent [55,56].

In recent years, there have been tremendous advances in conductor technology, nanomaterials, carbon nanotubes, and GR, which have had some impact on the quality of signals obtained from electrochemical biosensors and their process improvements. Thus, various biometric materials based on olfactory sensing elements are expected to eventually be used to construct more sensitive and ultra-selective nanobiosensors by integrating them with various nanomaterials [8]. Biosensors based on OR have great potential for development and have promising applications in numerous fields due to their high sensitivity and specificity. For example, they can be applied to drug discovery by detecting interactions between ORs and drugs, as well as detecting specific interactions between ORs and odor substances, providing a useful platform for basic olfactory research. However, this research is still at an early experimental stage, and commercial OR biosensors are not yet available on the market due to the fragility of the biosensing components and the lack of small, portable signal transduction systems. As we gain a better understanding of odor binding sites, synthetic proteins, and peptides with higher stability and reliability, these will likely replace tissues and cells for odor detection. In addition, microfabrication technology improvement will also accelerate the miniaturization of OR-based biosensors, and synthetic biology will likely facilitate their further development. Thus, with the development and combination of multiple disciplines, commercial OR biosensors are bound to emerge and show promising applications in many fields of application [31].

## 3. Commonly Used Signal Amplification Strategies for Electrochemical Biosensors

### 3.1. Signal Amplification Strategies Based on Nanomaterials

Nanomaterials are a type of material of an at least one-dimensional nanometer size (1–100 nm) in a three-dimensional space, and are characterized by their high electrical conductivity, good chemical stability, large specific surface area, and structural flexibility. Nanomaterials are endowed with unique surfaces, quantum size, and have been found to manifest macroscopic quantum tunneling effects [57], as well as having unique electronic and optical properties [58].

Nanomaterials allow direct contact with a sensing environment, enabling rapid signal conduction and thus increasing system sensitivity and reducing detection limits [59]. Nanomaterials have been commonly used as carriers or capture carriers to immobilize a large number of markers (e.g., antibodies, nucleic acids, and enzymes) based on their unique properties, such as their nanostructures or superparamagnetic activities [60]. In addition, nanomaterials have been used as novel luminescent reagents to enhance signals by modulating the luminescence of nanomaterials, such as by adjusting their size or ligands [61], thereby enabling signal amplification. Among the common nanomaterials, metal nanomaterials (e.g., gold and silver nanomaterials), carbon nanomaterials, quantum dots, and metal-organic frameworks can be directly used as electroactive substances to achieve signal amplification in sensors [62]. Table 1 lists some of the applications of nanomaterial-based signal amplification electrochemical biosensors for practical detection.

#### 3.1.1. Metallic Nanomaterials

Metal nanomaterials typically include metal nanoparticles (e.g., AuNPs, AgNPs, and PdNPs), metal oxide nanoparticles (e.g., CeO_2_ NPs and CuO NPs) polymetallic nanoparticles [78], and metal sulfide nanoparticles [79]. Metal nanoparticles have excellent electron transfer and biocatalytic capabilities, good electrical conductivity, biocompatibility, and chemical inertness [61,80]. In terms of signal amplification, metal nanoparticles can be used not only as electrode modifiers to improve biosensor sensitivity by providing a large active surface area and promoting electron transfer [81,82], but also can be used as redox tracers and catalytic markers to enable signal amplification [80]. Metal nanoparticles as catalytic markers have also been used to convert biometric events into significantly amplified signals by catalyzing specific chemical reactions. Compared to enzyme catalytic labeling, metal nanoparticles can overcome the limitations imposed by thermal stability and differential environmental stability [80].

Gold nanoparticles (AuNPs) have attracted attention due to their stability [83], ease of binding to biomolecules [84], affinity for mercaptan (-SH2) groups [85], and quantum effects related to shape and size [86]. In addition, AuNPs can form stable Au-S bonds with functional sulfhydryl groups, using functional groups containing these immobilized ligands as well as other parts of a ligand (e.g., nucleic acids or proteins) to eventually form functionalized AuNPs [59]. Adenosine triphosphate (ATP) is a signal molecule in processes such as photosynthesis, enzyme catalysis, and biosynthesis, and has been associated with many diseases, making it an extremely important nucleotide in living systems. For example, Li et al. [63] constructed a DNA sandwich structure through sulfhydryl chemistry and DNA self-assembly techniques, using gold electrodes as a substrate and nucleic acid aptamers as recognition elements. They used double-labeled methylene blue (MB) as an electrochemical probe, with signal amplification using gold nanoparticles and final quantification by differential pulse voltammetry. The detection principle of the above study is shown in Figure 3. A capture probe (CP), partially complementary to an ATP aptamer, was first immobilized on a gold electrode surface by gold–sulfur bonding and the residual active site on the gold surface was closed with 2-Mercaptoethanol MCH (MCH). Then, an MB-labeled DNA chain was assembled on the surface of these GNPs, and the signal chain was complementary to the remaining bases of the ATP aptamer. In this experimental design, both the ATP aptamer (SP1) and the complementary strand (SP2) contained MB, an electroactive molecule, which could enhance the electrochemical signals. In addition, these GNPs had a strong DNA immobilization capacity and could load multiple DNA chains containing MB, which further enhanced the electrochemical responsiveness and enabled signal amplification. Ochratoxin A (OTA), a secondary metabolite produced by Penicillin and Aspergillus, has been shown to be extremely important in food safety and human health monitoring because of its strong toxic, teratogenic, and carcinogenic effects [87]. Bai et al. [87] prepared a novel electrochemical nucleic acid aptamer sensor using gold nanostars (GNSs) and biomass nitrogen-doped porous carbon (NDPC) as dual signal amplification strategies, thus, realizing the efficient detection of OTA. Compared to other gold nanomaterials, GNSs have a multi-branched structure with a sharp tip, making them more suitable for the enrichment and detection of target molecules. Research has shown that doping nitrogen into porous carbon can improve its poor dispersion and enhance its electrical conductivity [88].

Silver nanoparticles (AgNPs) are also a widely used metal nanomaterial for electrochemical biosensors, and have the advantages of easy signal amplification, a high extinction coefficient, and a high ratio of scattered extinction [59]. Compared to AuNPs, AgNPs can be easily oxidized and produce significant amplification signals at low potentials. Hence, AgNPs have been often used as redox markers [89]. Dong [90] et al. used oligonucleotides encapsulated in silver nanoclusters (AgNCs) as probes, and these oligonucleotide probes were shown to have specific properties to mimic metalloenzyme-catalyzed H_2_O_2_ reduction due to their combination of recognition sequences for hybridization and stencil sequences for the in situ synthesis of AgNCs. In addition, copper nanoparticles have been widely used due to their minimal toxicity, good biocompatibility, and ease of synthesis.

Electrochemical biosensors based on metal nanomaterials have unique electrocatalytic activity compared to other conventional tools [91]. However, metal nanoparticles have been shown to be susceptible to salt concentration and exhibit problems of electrical instability. Furthermore, further research is still needed regarding the utilization of metal nanoparticles as carriers to ensure that diffusion limitations within a nanostructure do not affect individual biosensors. Currently, improving efficiency has been shown to be possible by applying appropriate surface modifications to metal nanoparticles to resist salt-induced aggregation and by determining the homogeneity of a nanomaterial, its distribution, and its shape [89,92].

#### 3.1.2. Carbon Nanomaterials

Carbon nanomaterials include carbon nanotubes (CNTs), graphene (GR), graphene oxide (GO), carbon nanofibers, and carbon quantum dots (CQDs). Compared to other materials, carbon nanomaterials have superior hardness, remarkable mechanical properties, better thermal and electrical conductivity than other nanomaterials [93], and they are often used in electrochemical biosensors.

Carbon nanotubes, including single-walled CNTs (SWCNTs) and multi-walled CNTs (MWCNTs), have remarkable one-dimensional electrical conductivity, inherent hollow structures, and can adsorb specific biomolecules on their surfaces. As a result, CNTs have been used as labeling materials and can be loaded with a large number of biomolecules, allowing for signal amplification. Microcystin-LR (MC-LR) is one of the most toxic known cyanotoxins, and its degradation intermediate linear microcystin (L-MC-LR) has also been shown to be quite toxic in humans [94]. Thus, microcystins can significantly pollute water resources and pose a threat to human health. In contrast to previous studies that have used antigens, antibodies, and molecularly imprinted polymers (MIP) as receptors, Li et al. [94] innovatively used the degradation enzyme MIrB as a recognition element that specifically bound to the target molecule MC-LR. In this study, the diverse surface chemistry and super-electric activity of MWCNTs were exploited to modify an electrode with the -COOH functionalized MWCNTs to increase the specific surface area of this electrode while facilitating the immobilization of incoming MIrB to achieve signal amplification. Recently, the problem of pesticide residue and exceedances in food has become a major global public health threat. As a result, it has received significant worldwide attention from the research community. Liu et al. [64] constructed an electrochemical biosensor for the accurate detection of organophosphorus pesticide dichlorvos using wheat esterase as a detection enzyme source and synergistic electrocatalytic amplification of AuNPs and MWCNTs. This research has provided good technical support for pesticide detection.

GR and GO have been shown to have great potential as biosensors due to their unique electronic and fluorescent properties, remarkable electrical conductivity, large specific surface area, and strong mechanical strength [95]. In electrochemical enzyme biosensors, GR was shown to not only connect an electrode surface to an enzyme to achieve electron transfer, but was also shown to enhance the electrochemical signal response [96]. In addition, GR and its related materials can be used as a direct marker in immunoassays using a working signal generated by the reduction of its oxygen-containing groups, or they can rely on the modification of this working signal with other electroactive probes to generate an indirect marker [97,98]. Bonanni et al. [68] constructed an electrochemical biosensor using GO as a direct marker for the detection of DNA polymorphisms. These researchers fixed a DNA probe on a carbon electrode, hybridized it with a single nucleotide polymorphic DNA sequence, and used a non-complementary sequence as a negative control. The irrational use of organophosphorus and its residues in agricultural products can cause harm to humans, such as causing gastrointestinal disorders and aggravating the burden on the liver. In general, organophosphorus pesticides inhibit the activity of acetylcholinesterase (AChE). Liu et al. [69] designed a DNA signal amplification electrochemical biosensor based on laser-induced GR (LIG) and MnO_2_ switch-bridging for the rapid detection of pesticides. Compared to traditional GR, laser-induced GR (LIG) was shown to be a cost-effective and environmentally friendly tool, and its morphology could also be controlled [99]. These researchers integrated LIG electrodes on polyimide films and loaded MnO_2_ nanosheets on paper to construct a novel electrochemical biosensor for the detection of organophosphorus (OPs) residues. These researchers also used acetylcholinesterase-catalyzed hydrolysates to trigger the cleavage of an MnO_2_ nanosheet layer, thereby releasing DNA, which in turn initiated nicking enzyme cyclic amplification and, ultimately, signal amplification.

Electrochemical biosensors based on carbon nanomaterials offer the advantages of having excellent physical and electrical properties, low cost, and ease of handling. However, it is difficult to control the impurities and substrates of GR, as well as the diameter, chirality and degree of agglomeration of CNTs [100]. Thus, the performance of electrochemical biosensors based on carbon nanomaterials has not been found to be consistent. Currently, the development of electrochemical biosensors based on carbon nanomaterials is still in its initial stages and faces many challenges and opportunities; thus, corresponding efforts should be made to improve their reproducibility, biocompatibility, and nanotoxicity, and to promote their development in a wider range of applications [89].

#### 3.1.3. Quantum Dots

Quantum dots (QDs) are fluorescent semiconductor nanostructures with a particle size between 1–10 nm, usually composed of elements of groups II–VI or III–V, and are also referred to as nanocrystals [101,102]. QDs have good photochemical and optoelectronic properties [103,104], especially due to their high electron transfer efficiency and good surface reactivity [59], and they have been widely used as ideal markers in electrochemical biosensors. QDs can generally be divided into two categories, namely, sulfur-compound QDs and carbon-based QDs [105]. Sulfur-compound QDs, such as cadmium sulfide QDs (CdS), cadmium telluride QDs (CdTe), and lead sulfide QDs (PbS), are the most typical semiconductor QDs that are often used to build electrochemiluminescent sensors (ECLs). Graphene QDs (GQDs) are a type of carbon-based QD. Compared to GR, GQDs are smaller, more stable in terms of chemical properties and electrical conductivity, and are rich in carboxyl groups [106], making them easy to functionalize. Wang [70] et al. constructed a sensitive electrochemical biosensor for the detection of miRNA-155 based on GQDs and horseradish peroxidase (HRP). These researchers leveraged the advantages of GQDs such as their good biocompatibility and high conductivity as carriers for immobilized enzymes, and immobilized a large amount of HRP on GQDs while using the catalytic effect of HRP to finally achieve signal amplification. The detection limit of this electrochemical biosensor was 0.14 fM.

#### 3.1.4. Magnetic Nanoparticles

Magnetic nanoparticles (MNPs) are nanoscale particles that usually consist of a magnetic core made of metal oxides such as iron or cobalt (e.g., Fe_3_O_4_) and a polymer wrapped around the outside of this magnetic core. As such, they are considered a new class of synthetic materials. Magnetic nanoparticles have excellent physicochemical properties, high stability, as well as a low synthesis cost, and electrochemical sensors can improve their sensitivity and shorten detection time through the use of magnetic nanoparticles [107].

Magnetic nanoparticles have two signal amplification principles. First, due to their high electrical conductivity and large specific surface area, magnetic nanomaterials have often been used as core materials in synthetic molecular polymers to enrich and capture target molecules in complex environments, thereby, enhancing an electrochemical signal. Second, they have been used as electrode surface modifiers to amplify the electrochemical signals by increasing the rate of electron transfer [108]. 17-b-Estradiol is a natural steroidal estrogen whose concentration affects the human endocrine system and consequently human health [109,110]. Molecularly imprinted polymers (MagMIPs) modified with magnetic nanoparticles are highly selective and sensitive compared to normal molecularly imprinted polymers (MIPs), allowing for rapid and effective binding to a target [111,112]. Lahcen et al. [73] developed a new method of surface modification of screen-printed carbon electrodes (SPCE) using magnetic molecularly imprinted polymer nanocomposites (Fe_3_O_4_-MIP) with surface modification to construct an electrochemical biosensor for the detection of 17-b-estradiol. They used these Fe_3_O_4_ NPs to increase the surface area of the sensor, thus enhancing the detected oxidation current signal.

Magnetic nanoparticles offer the advantages of paramagnetism, biocompatibility, and ease of surface modification [74]. In addition, they have the advantage of mimicking enzymes and nano-electrocatalysts [113]. Electrochemical biosensors based on magnetic nanoparticles not only combine the advantages of magnetic separation and enrichment, but also provide additional analytical detection capabilities based on magnetic properties, thus significantly improving operational convenience and detection performance [113]. As a result, they have been widely used in practice for the analytical detection of targets.

#### 3.1.5. Metal-Organic Framework Materials

Metal-organic framework materials (MOFs) are a class of organic-inorganic hybrid materials formed by the combination of metal ions or clusters and organic ligands as linkers. The structure and function of MOFs formed by the combination of different metals and connecting groups differ. MOFs can be characterized by their porous nature, large specific surface area, and high thermal stability [59]. In addition, MOFs have been shown to be well-suited for loading bioligand molecules based on various functional groups (e.g., NH_2_, -COOH), pi–pi stacking, and other interactions [76,114]. In biosensors, MOFs not only provide high loading of probe DNA, but have also shown to be resistant to the degradation of probe DNA [115]. Conventional pure MOFs have been shown to be chemically less stable and decompose at high temperatures [116]. Therefore, many researchers have proposed combining MOFs with functional materials to form novel MOF composites. For example, using functional materials such as metal nanoparticles, enzymes, and QDs to modify MOFs, the constructed composites would not only overcome the drawbacks of pure MOFs, but also combine the excellent properties of both materials [59,116].

Lei et al. [116] combined GR with MOFs based on GR’s advantages of high electrical conductivity and electrocatalytic activity to construct a GR/MOF composite electrochemical biosensor. These researchers combined the excellent electrical, thermal, and optical properties of GR with the unique properties of the MOFs, exploiting the synergistic effect of both to significantly amplify an electrochemical signal. Acetaminophen (PA) is a drug that can be used to lower body temperature and relieve pain [117], while dopamine (DA) is a neurotransmitter in the brain that controls a variety of physiological functions in the human body. Generally, PA and DA will be present in organisms simultaneously and interfere with each other [118]. Ma et al. [76] constructed an electrochemical biosensor based on HKUST-1 (Cu-BTC) coupled with GO (ERGO) for the simultaneous detection of PA and DA. These researchers deposited composites on glassy carbon electrodes and reduced GO to graphene oxide (ERGO) to exploit the synergistic effect of the conductivity of ERGO and the porous structure of MOF to achieve signal amplification.

Electrochemical biosensors based on composites of MOFs have the advantages of both, with the high porosity and ordered porous properties of the MOFs and the electrochemical and catalytic properties of the other functional materials [119]. However, MOFs are still at an early stage of research in the field of sensors, especially in the field of electrochemical sensors, and the problem of the poor electrical conductivity and stability of MOFs has made their development extremely challenging [120]. Therefore, in future work, it will be necessary to further explore the conformational relationship between the reaction condition-structure-electrochemical performance of MOFs and to construct MOFs with efficient electroactivity and electrocatalytic activities. Thus, novel electrochemical biosensors with excellent sensitivity, stability, selectivity, and accuracy could be prepared, broadening the potential of MOF-based biosensors in the fields of bioanalysis, clinical medicine, and analytical chemistry [121].

### 3.2. Signal Amplification Strategies Based on Enzymes

Enzymes are an important class of biocatalysts that enable the determination of very small amounts of targets due to their high catalytic efficiency, strong specificity, and high sensitivity [60], easing the detection of currents generated by the redox reactions of enzyme-catalyzed substrates in enzymatic reactions [23]. The main methods of enzyme immobilization include physical adsorption, covalent binding, encapsulation, and cross-linking [121]. Enzyme biosensors consist of immobilized enzymes and signal transducers that detect the presence of a specific analyte by measuring the changes in proton concentration (H+), the release or absorption of gases (e.g., CO_2_, NH_3_, or O_2_, among others), light absorption or reflection, and thermal radiation that occurs during substrate consumption or enzyme reaction product formation. These sensors can then convert these into a measurable signal (electrical, optical, or thermal) [122,123] through which the analyte to be measured can be indirectly measured [121].

Enzymes have also been often used as signal markers in biosensors, not only for simultaneous determination using multiple enzymes, but also for signal amplification using several different coupling techniques. From an economic point of view, the experimental costs can be generally reduced by obtaining inexpensive experimental reagents [124]. Common enzyme labels include HRP, alkaline phosphatase (ALP), glucose oxidase (GOX), cytochrome peroxidase, and DT-diaphorase (DT-D). Kang et al. [125] constructed an electrolyte based on DT-diaphorase (DT-D) as an oxidoreductase label and 4-nitroso-1-naphthol (4-NO-1-N) as a substrate. They used DT-diaphorase (DT-D) to convert electrochemically inactive nitroso compounds into electrochemically active amines that can participate in redox cycling reactions. Zhao et al. [126] prepared an electrochemical biosensor based on the display of tyrosinase on the cell surface of *E. coli* for the detection of bisphenol A (BPA). The lower detection limit of this biosensor was 0.01 nm at BPA concentrations ranging from 0.01 to 100 nm. This immobilization of the enzyme on the cell surface not only reduced the loss of enzyme activity, but also facilitated regeneration of the cellular enzyme system.

Enzyme-based electrochemical biosensors are small, simple to operate, and economical, and have the advantages of real-time monitoring capability, high sensitivity, and strong specificity. However, the construction of enzyme electrochemical biosensors remains challenging due to the complex protein structures of enzymes. In addition, their active centers are often hidden inside a structure, where direct electron transfer between an enzyme and an electrode can often be difficult, and the environment on the electrode surface can also lead to reduced enzyme activity [127,128]. Currently, the molecular structures of enzymes can be modified by techniques such as chemical modification, recombinant enzyme molecules, and targeted mutagenesis [129,130,131], or by adjusting the modification process of enzyme molecules on the electrode surface to maintain the enzyme activity and expose the active center, thus improving the efficiency of the direct electron transfer of enzyme molecules [132].

### 3.3. Signal Amplification Strategies Based on Nucleic Acid Amplification Techniques

#### 3.3.1. Signal Amplification Strategies Based on Nuclease

Nucleic acid amplification is a process for the efficient amplification of specific nucleic acid sequences and it has been often used to detect a variety of biological targets (e.g., DNA, RNA, and other small biological molecules). There are two types of target cycling principles in nucleic acid amplification, one of which involves the copying of the target chain by nucleic acid amplification, resulting in signal amplification, while the other involves a nuclease-induced target cycling-based strategy, which has been widely used in signal amplification for electrochemical biosensors. The various nuclease tool enzymes that have been often relied upon for nucleic acid amplification based on nuclease assistance consist of nucleic acid endonucleases, nucleic acid exonucleases, polymerases, and DNAzyme [59]. Double-strand-specific nucleases (DSN) are the most commonly used nucleic acid endonucleases [133]. Polymerase chain reaction (PCR), rolling circle amplification (RCA), and strand displacement amplification (SDA) are all common polymerase-dependent nucleic acid amplification reactions. PCR is a target amplification technique, and the detection principle of RCA technology involves the following. It uses a circular single-stranded DNA as a template, and under the drive of DNA polymerase, a large amount of single-stranded DNA complementary to this circular DNA template be produced, which finally amplifies a signal [59]. In addition, RCA products have been often used as carriers loaded with large amounts of electrochemical tracers to achieve electrochemical signal amplification [134,135].

Yang et al. [136] first used circular enzyme signal amplification (CESA), DSN, and 3-QD-DNA nano-complexes as cascade signal probes to achieve the ultrasensitive detection of microRNAs using dual signal amplification. Miao et al. [137] prepared an electrochemical biosensor for miRNA detection based on the catalytic reaction of double-stranded specific nuclease (DSN) and cleaved endonuclease (NEase). These researchers designed a DNA four-way linkage structure on an electrode surface, and with the assistance of a DSN cleavage reaction and enriched DNA probe that could be triggered by a target miRNA, the electrochemical material on the electrode surface was heavily consumed after the cyclic catalytic reaction. In addition, the determination of target miRNA concentration could be achieved based on the electrochemical response. Zhang et al. [138] designed an electrochemical sensor based on RCA-mediated palladium nanoparticles (PdNPs), achieving the ultrasensitive detection of microRNAs.

DSN-based signal amplification strategies have been one of the most effective ways to improve the sensitivity and specificity of electrochemical biosensor detection, and have shown good potential in the early diagnosis of diseases. However, DSN-based electrochemical biosensors still face significant challenges in the design of sensing probes. In addition, there are still many unresolved issues in terms of the standardized methods for high selectivity and sensitivity in the detection of actual clinical samples [133].

#### 3.3.2. Signal Amplification Strategies Based on Enzyme-Free Nucleic Acids

The strand displacement reaction (SDR) is a common enzyme-free nucleic acid amplification technique, and in recent years, researchers have proposed a sticky terminal-mediated strand displacement reaction (TMSDR), which has greatly accelerated the reaction speed of SDR [59]. TMSDR amplification techniques have been widely used in biosensors due to their advantages such as being enzyme-free, their high amplification efficiency, and their controlled kinetics [59]. Both common nucleic acid amplification techniques, catalytic hairpin self-assembly (CHA), and the hybrid chain reaction (HCR) fall under the category of TMSDR. Catalytic hairpin self-assembly (CHA) achieves signal amplification by using the catalytic function of an initiator chain to release the initiator chain after forming a double-stranded structure, proceeding to the next round of catalytic hairpin self-assembly [139]. The hybrid chain reaction (HCR) consists of a probe amplification technique that uses molecular recognition and hybridization reactions to sequentially open multiple hairpin probes to achieve signal amplification through cumulative signals [140,141]. The amplification principle of HCR is shown in Figure 4, where the two sub-stable DNA hairpins (H1 and H2) complementary to each other each have a short sticky end, but are bound by the hairpin structure and can only be amplified through the addition of an initiating chain (I) and a toehold chain replacement reaction with H1, where H1 opens and a new single-stranded domain is exported to react with H2. Then, H2 opens, and a new single-stranded domain is similarly exported to react with H1. This reaction follows a chain reaction that culminates in the formation of a long dsDNA polymer [142].

Ling et al. [143] constructed an electrochemical biosensor for efficient protein detection based on the molecular recognition between an aptamer and a target utilizing a target departure signal amplification strategy. They hybridized the auxiliary DNA1 with an aptamer, forming a double-strand, and then used the high affinity between the aptamer and target to release the auxiliary DNA1 and trigger the DNA1 circuit, eventually forming a large number of hairpin-shaped DNA3 on the electrode surface. Thus, the unfolded DNA hybridized with the DNA captured on the surface of the Pt nanoparticles would generate an electrochemical signal of H_2_O_2_ reduction for signal amplification. Cheng et al. [144] constructed an electrochemical biosensor for the efficient detection of exosomal microRNAs based on the strand displacement reaction (SDR) strategy of signal amplification of both target miRNA cycling and silver nanoparticle deposition. The detection limit of this electrochemical biosensor was shown to be 0.4 fM (S/N = 3) for miRNA-21 in exosomes [144].

CHA has the advantages of high catalytic efficiency, low background signal, and a simple and stable reaction system. The use of HCR can ensure both the sensitivity and specificity of experiments, and compared to traditional methods, HCR has been shown to be an isothermal reaction process that can be performed at room temperature, yielding a high signal-to-noise ratio [133]. Based on this signal amplification strategy, electrochemical biosensors as a detection platform for the highly sensitive detection of targets with fewer restrictions on the experimental environment provide a simple yet versatile method, with promising applications for basic research and clinical detection. However, there are still challenges, such as the ability to effectively control background signals in the analytical application of CHA, and the limitations of the HCR in terms of a low sensitivity to DNA linker formation and a low catalytic rate [145].

### 3.4. Signal Amplification Strategies Based on Polymers

With the development of modern technology, complex and optimized soft structures with synergistic properties have been developed to improve the sensitivity and selectivity of existing biosensors [146]. Polymers such as polyaniline (PANI), polypyrrole, and transition metal oxides have been widely used in biosensors due to their good electrical conductivity, ease of compounding with other functional materials, and low cost [60,147,148,149]. In addition, they can further enhance electrochemical signals by accelerating electron conductivity [91]. Transition metal sulfides, with their high electrical conductivity, large active surface area, and high catalytic activity, have also been shown to be suitable and desirable materials [150,151,152]. Covalent-organic frameworks (COFs) are a new polymer consisting of precisely organized organic building units with a periodic skeleton and porous crystal structure connected using covalent bonds [153]. In addition, this polymer can be used in combination with other materials (e.g., metal nanomaterials and carbon nanomaterials) to synergistically generate a large number of signals by combining the strengths of both materials, thus enabling signal amplification.

Polydopamine (PDA) is a good conductive polymer coating material that can be very easily deposited on various material surfaces [154]. Zheng et al. [153] first designed an electrochemical biosensor using polydopamine (PDA)-coated BCN as a substrate platform, methylene blue (MB) containing a MnO_2_-functionalized COF as a signal amplification platform and probe material, and metallic gold-platinum nanoparticles (AuPbNPs) for signal double amplification of the electrochemical biosensor to achieve the ultra-sensitive detection of PSA. These researchers used the synergistic behavior of MnO_2_ nanoparticles to ensure that the COF did not aggregate in this reaction, thus ensuring the stability of the composite. Molybdenum sulfide (MoS_2_) is a transition metal sulfide with a large specific surface area, high catalytic properties, and good optical properties. However, it is often functionalized in electrochemical biosensors due to its lack of conductivity [155,156,157]. Sun et al. [158] assembled an electrochemiluminescent immunosensor based on AMGM nanocomposites for the detection of PSA in serum using a so-called immunosandwich method. They prepared the MoS_2_/GO/o-MWNT nanocomposites with a three-dimensional flower-like structure using GR and CNTs as supporting skeletons, which were then modified with gold nanoparticles using a hot water method. The polymer amplified signal amplification by increasing antibody loading, accelerating the rate of electron transfer, and maximizing the retention of antibody activity. The ECL response of this electrochemiluminescent immunosensor was proportional to the logarithm of the PSA concentration, and had a detection limit of 0.1 pg/mL. Molecularly imprinted polymers (MIPs) and surface imprinted polymers (SIPs) have been shown to have stable physicochemical properties and have often been used in the construction of electrochemical biosensors [159].

Polymer-based electrochemical biosensors have provided great improvements in sensitivity, selectivity, stability, and the corresponding reproducibility of electrodes for the detection of a wide range of analytes [160]. Due to their having a large specific surface area and good biocompatibility, polymers can be used in combination with other materials to take advantage of both characteristics, and ultimately produce a large number of signals. Thus, polymers provide researchers with a way to prepare efficient biosensors when used in combination with other materials [159].

### 3.5. Signal Amplification Strategies Based on Redox Markers

In microRNA electrochemical biosensors, redox markers have typically been used to enhance the electrical signals by electrostatic aggregation on adsorbed electrode surface microRNAs or to generate heterologous double-stranded nucleic acid molecules to capture DNA/RNA for signal enhancement [133]. Methylene blue (MB), toluidine blue (TB), and hexaammonium ruthenium (RuHex) are all common redox signal indicators used in electrochemical biosensors. Tian et al. [161] constructed a simple and sensitive electrochemical biosensor for the detection of microRNA-21 by modifying an AuNP superlattice onto a glassy carbon electrode for the first time, enhancing the specific surface area and conductivity of the electrode and using the electrostatic adsorption aggregation of TB for signal amplification. Hexaammonium ruthenium (RuHex) is a positively charged electroactive complex that binds to the anion of a deoxyribonucleic acid through electrostatic interactions. Hong et al. [162] constructed an efficient electrochemical biosensor for microRNA detection using RuHex as a signal indicator and screen-printed gold electrodes (SPGEs) as a substrate. These researchers introduced two auxiliary probes self-assembled as signal amplification carriers, and when the target molecule was present, the DNA conjugate formed by the auxiliary probe self-assembly could bond to the capture probe, and then RuHex aggregated through the DNA conjugate to the working electrode, ultimately achieving signal amplification [162].

Redox molecules can bind directly and specifically to DNA without complex preparation. Therefore, direct intercalation of redox molecules can be considered a simple and rapid intercalation method. However, electrochemical biosensors based on redox labeling cannot simultaneously detect multiple substances, as the interaction of redox molecules is not specific and can be embedded on all DNA chains on an electrode surface, leading to non-specific interactions and ultimately affecting the detection results [133].

### 3.6. Signal Amplification Strategies Based on Cells or Tissue

Traditionally, tissue sensors are a class of biosensors that use animal and plant tissue-thin layer slices as receptors for signal amplification by utilizing the catalytic action of enzymes in natural tissues. Microbial biosensors are composed of molecular recognition elements (microbial sensitive membranes) and signal transducers. A microbial sensitive membrane can be fabricated by applying immobilization techniques to immobilize microorganisms onto a carrier without damaging their function. The principle of operation involves using respiration or metabolism for signal amplification purposes. In recent years, studies have shown that cells or tissues can be used directly as sensitive elements to construct corresponding electrochemical biosensors for the detection of targets and for signal amplification [163].

Cells can sense small changes in environmental signals through receptor–ligand interactions, and these signals can be amplified through a cellular system of signaling cascades, which can cause changes in membrane potential and depolarization by regulating ion channel switches; thus converting signals into neural signals that can be transmitted to brain nerve centers to produce rapid responses [164]. In addition, these signals can be transmitted to the nucleus via cellular signal amplification pathways, and then signals such as metabolism are sent throughout the body by the regulation of gene expression, thus allowing the system to respond to environmental factors [165]. One of the primary components of this process is the G-protein cascade amplification system, which is coupled with multiple receptors to sense and amplify cellular signals [164]. To explore whether GPCRs can trigger G-protein signal amplification in tissues or cells of different species, Xu et al. [164] self-assembled bombykol receptors onto the cell membranes of taste bud tissues of different species. Eventually, a novel bombykol receptor sensor was established to detect G-protein signal amplification. The results showed that the G-protein signal cascade amplification system was universal in the GPCR of different tissues and species. Among them, the bombykol receptor sensor self-assembled with the cattle taste tissue, which was the most sensitive toward bombykol, with a lower limit of detection of about 1 × 10^−19^ mol/L.

Wei et al. [166] constructed an electrochemical taste biosensor by simulating the neurotransmission mechanism of taste, using porcine taste bud tissue as the sensitive element to measure the effects of bitter taste receptors interacting with sucrose octaacetate, benzoate, and quercetin. The results showed that this sensor could be used to characterize cellular or tissue receptor–ligand interactions and its cellular signal cascade amplification. In addition, the signal amplification of the taste bud cells was more than ten orders of magnitude. Resveratrol is a polyphenol phytoalexin found in a variety of plants that has been shown to have a wide range of health benefits in animal studies [167,168,169,170] and beneficial biopharmacological activity against cancer in humans [171]. However, a limited number of studies have evaluated its interactions with cell surface receptors. Ren et al. [172] constructed a sandwich rat small intestinal tissue sensor (RSIT sensor) based on rat small intestinal tissue cells as the sensitive elements and effectors to detect the corresponding currents of different concentrations of resveratrol for receptor stimulation by electrochemical means, and compared the response values of this electrochemical biosensor and the bare electrodes prepared in this study with those of resveratrol. The results showed that the lower limit of detection was 1 × 10^−13^ mol/L, and the intracellular signaling system significantly amplified the response values of the small intestinal cells to resveratrol, by approximately 100 times.

Electrochemical biosensors based on cells and tissues have shown good stability and accuracy and have broad application prospects for pheromone detection, sex pheromone detection, and receptor structure and function. However, further research is needed to determine whether the immobilization of cells affects receptor–ligand interactions.

### 3.7. Signal Amplification Strategies Based on Microfluidics

Microfluidics refers to the science and technology of precisely controlling and manipulating micro and nanofluids with high precision and reproducibility in the micro and nanoscale space [173,174]. Microfluidic chips, also known as lab-on-a-chip (LOC), can integrate the basic functional units involved in biological and chemical experiments, such as mixing, reaction, separation and detection, on a single chip with a high degree of integration [175]. In addition, the microfluidic chip is a critical part of the biosensor for microfluidic control technology, and its structure is shown in Figure 5. It mainly consists of three parts: a micro-mixer with staggered asymmetric herringbone recesses, a serpentine culture channel, and a separation chamber [176]. As a key component of the microfluidic chip, the micromixer often has a great impact on the sensitivity of the microfluidic biosensor. The top or bottom of the microchannel has been shown to promote lateral flow, increase local vorticity and achieve effective mixing [177,178].

In recent years, there has been an increasing awareness of health and a growing need for the prevention and diagnosis of several diseases. The combination of microfluidics and advanced biosensing technologies has given rise to many excellent miniaturized analytical platforms that enable the precise control of micro- and nano-fluids and the integration of various types of bio arrays on a miniaturized platform [174]. Microfluidics, due to its rapid development, has gradually become the main means of implementation for POCT diagnostics. Electrochemical biosensors based on microfluidic control technology have been frequently applied for the monitoring of cells and cellular components, proteins and small molecules, etc. [179]. Hazal Kutluk et al. [180] constructed a low-cost microfluidic biosensor for the electrochemical measurement of miRNA-197 (a tumor biomarker candidate) in undiluted human serum samples, which required a very low sample volume (580 nl) and short detection time. They developed two different formats (sandwich assay and competition assay) for miRNA determination and compared them in terms of sensitivity, accuracy and simplicity. The results showed that the sandwich assay had a superior performance in terms of sensitivity and selectivity with a minimum detection limit of 1.28 nM. Cardiac troponin I (cTnI) is an attractive biomarker for acute myocardial infarction (AMI). Yang L et al. [181] developed a sensitive portable microfluidic electrochemical array device (μFED) for the immunoassay of trace human cardiac troponin I (cTnI). They used an alkaline phosphatase (AP) label for signal amplification and improved the performance of the μFED by eliminating the shielding effect of the microelectrode array (MEA) integrated with the μFED. The results showed that the detection time for cTnI was only 15 min and the lower limit of detection was 5 pg/mL (S/N = 3).

In summary, microfluidic integrated biosensor devices have many advantages, such as low reagent consumption, short reaction time, automated sample preparation, high throughput analysis, high detection accuracy and portability, and low cost, and are often applied to the detection of various substances [182,183]. However, most of the current microfluidic products are limited to the laboratory stage of scientific research, and there are few mature products on the market. Moreover, the application of microfluidics is limited by the ultra-high precision processing requirements, the challenge of precise control of micro- and nano-sized liquids, and the problem of how to achieve rapid mass production at a low cost. In recent years, the introduction of various functional materials has facilitated the development of multifunctional microfluidic control chips, which have broadened the application areas of this type of biosensor and are expected to shine in the biomedical field [174].

### 3.8. Signal Amplification Strategy for the Combination of Multiple Materials

In recent years, to achieve the amplification of electrical signals, researchers have moved away from using a single material and instead have designed signal amplification strategies based on the combined use of multiple materials, each of which has its advantages in various detection methods. These include metal nanomaterials for their high electrical conductivity and electron transfer capability, carbon nanomaterials for their large specific surface area, high electrical conductivity, and physical-chemical stability, and enzymes for their high catalytic properties and specificity. For example, Zhang et al. [184] constructed a highly sensitive electrochemical biosensor for microRNA detection based on three signal amplification strategies, namely double-stranded specific nuclease (DSN)-assisted target cycling, gold nanoparticles (AuNPs), and HRP enzyme-catalyzed reactions. These researchers combined excellent enzymatic activity with a large specific surface area and better controlled the surface properties of gold nanoparticles [90,185], as well as using the strong differentiation ability of DSNs to achieve signal amplification with high sensitivity and selectivity. Chen [186] et al. also developed an electrochemical DNA biosensor combining three signal amplification techniques using the rolling circle amplification (RCA) shear recovery of target DNA, gold nanoparticle labeling, and multiple probes for the detection of MON89788 in soybean transgenic gene sequences. This electrochemical biosensor has a detection limit of 4.5 × 10^−17^ mol/L, and it has been often used for targeted gene sequence analysis because of its high sensitivity. Ferrocene (FC) has often been used as a signaling marker molecule due to its active reversible redox electron and excellent electrochemical signaling ability. Zhang et al. [187] used synthesized bis (ferrocene-labeled) hairpin DNA (placed at both ends of hairpin DNA) and integrated it into an electrochemical biosensor together with catalytic hairpin assembly (CHA) to achieve the ultra-sensitive detection of a microRNA, using the dual signal amplification of both. The synthesized double ferrocene produced two ferrocenes, which enhanced the signal response through their superposition effect, and this electrochemical biosensor had a minimum detection limit of 0.1 fp. Table 2 lists some applications of electrochemical biosensors based on the above signal amplification in practical detection.

## 4. Summary and Outlook

In recent years, with the continuous development of science and technology, such as materials, electronics, signal processing technique and gene editing technologies, signal amplification strategies based on electrochemical biosensors have made great progress and have become widely used. In this review, we researched and presented the progress on olfactory and taste electrochemical signal amplification strategies. Taste receptors have very important neurological, physiological, immunological, and endocrine functions in the presence of taste components [34]. In addition, we summarized various methods for enhancing electrochemical signals. Gold nanomaterials, carbon nanomaterial QDs, and other nanomaterials and enzymes commonly used as electrode materials, redox tracers catalytic markers, and carriers of signal elements [80], as well as signal amplification using nucleic acid amplification techniques have played an important role in enhancing sensitivity and improving the performance of electrochemical biosensors.

Electrochemical biosensors based on the above-mentioned signal amplification strategies will undoubtedly offer great potential for practical sample detection and olfactory taste determination in humans and animals. Currently, recent achievements in biotechnology have led to the possibility of modulating the affinity of bioreceptors for selected VOCs by gene editing techniques to design highly selective biosensors [8]. Furthermore, OR-based biosensors have great potential to become bioelectronic snuff systems for detecting VOCs in areas including food safety, environmental and industrial monitoring, clinical diagnostics, agricultural diseases, and drug development [31]. However, current electrochemical sensors still have some shortcomings. For example, the instability of their characteristics and final analytical performance between sensors in practical applications may lead to experimental failures [189]. Therefore, the development of electrochemical biosensors with high accuracy and stability that are suitable for general promotion and use still requires continuous effort to promote the development of electrochemical biosensors toward functional diversification, miniaturization, and integration, offering more extensive application prospects for disease diagnosis, genetic testing and other fields [127]. We believe that competitive research and large commercial opportunities in the field of biosensors will lead to exciting new developments in the near future.

## Figures and Tables

**Figure 1 biosensors-12-00566-f001:**
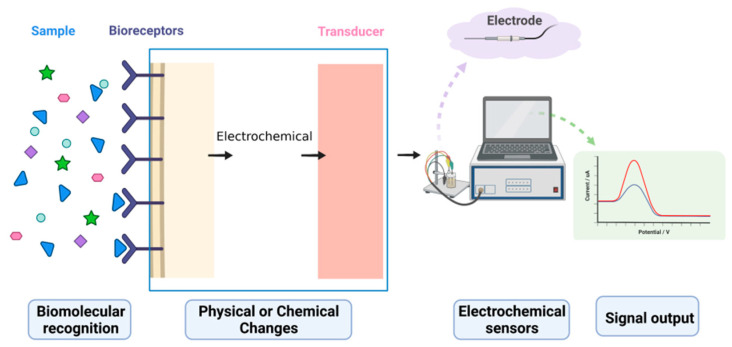
Basic principles of electrochemical biosensors.

**Figure 2 biosensors-12-00566-f002:**
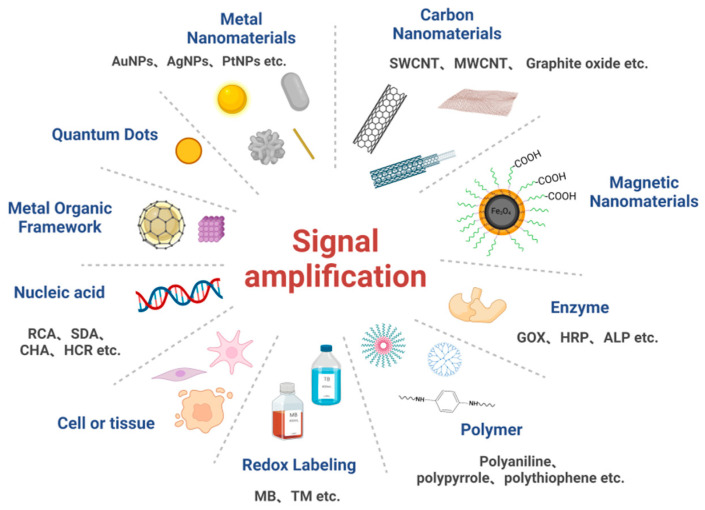
Signal amplification strategies commonly used in electrochemical biosensors.

**Figure 3 biosensors-12-00566-f003:**
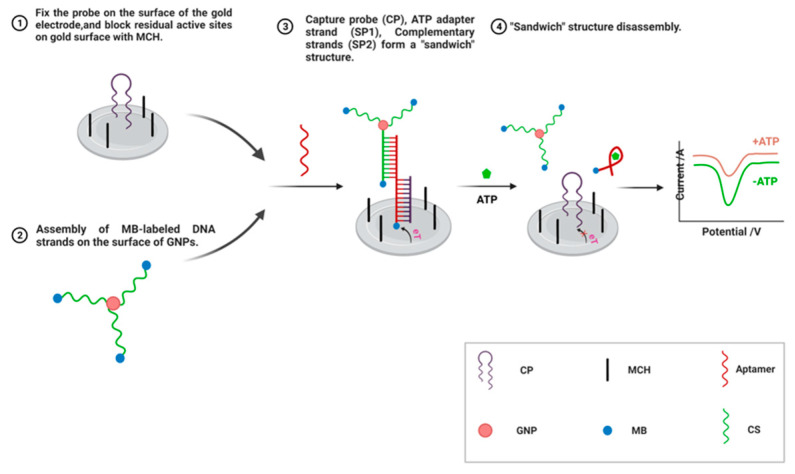
Principle of ATP electrochemical biosensor detection based on bio-nano assembly and signal amplification.

**Figure 4 biosensors-12-00566-f004:**
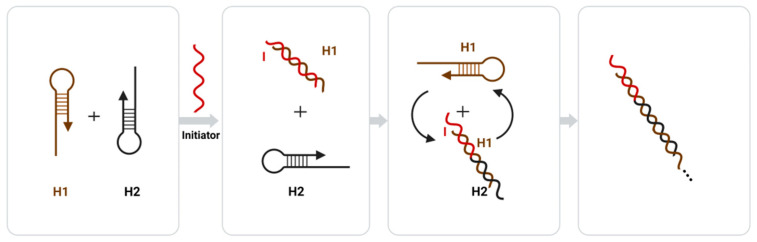
Principle diagram of the hybridization chain reaction (HCR) amplification.

**Figure 5 biosensors-12-00566-f005:**
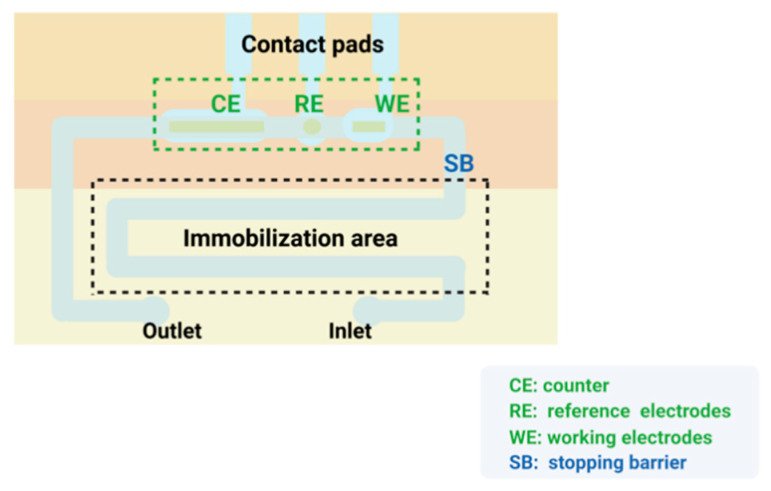
The structure of microfluidic chip.

**Table 1 biosensors-12-00566-t001:** Nanomaterial-based signal amplification strategies.

Strategies	Examples	Limit of Detections	Linearity Ranges	Ref.
Metallic nanomaterials	Electrochemical aptasensors for ATP detection based on sulfhydryl chemistry and DNA self-assembly techniques and gold nanoparticles	29.6 aM	ATP:10 fmol/L–1 mmol/L	[63]
	Electrochemical biosensor based on gold nanoparticles and multi-walled carbon nanotubes for the detection of dichlorvos	4 μg/L	10–100 μg/L	[64]
	Reusable miRNA biosensor based on electrocatalytic properties of heterogeneous double template copper nanoclusters (CuNCs)	8.2 fM	25–300 fM	[65]
	Detection of lipopolysaccharide by aptasensor based on gold cluster	7.94 × 10^−3^ amol/L	0.01 amol/L–1 × 10^−6^ amol/L	[66]
Carbon nanomaterials	MIrB is used as a recognition element, and the electrode modified with -COOH functionalized MWCNT to detect microcystin-LR	0.127 pg/mL	1 pg/mL–100 ng/mL	[67]
	Electrochemical biosensor using graphene oxide (GO) as a direct marker for the detection of DNA polymorphs	-	OTA:310 fM–310 pM	[68]
	Based on laser-induced graphene and MnO_2_ switch-bridged DNA signal amplification for sensitive detection of pesticides	1.2 ng/mL	OPs: 3–4000 ng/mL	[69]
Quantum Dots	Electrochemical biosensor for detection of miRNA-155 based on graphene quantum dots and horseradish peroxidase (HRP)	0.14 fM	miRNA-155: 1 fM–100 pM	[70]
	Detection of Alzheimer’s disease biomarker ApoE by electrochemical biosensor based on cadmium-selenium/zinc sulfide quantum dots	~12.5 ng/m L	10–200 ng/m L	[71]
	An electrochemical aptasensor to detect epithelial cell adhesion molecules (EpCAM) using silica nanoparticles and quantum dots	10 amol/L	10 amol/L–1.0 × 10^8^ amol/L	[72]
Magnetic nanoparticles	An electrochemical biosensor to detect 17-b-estradiol using magnetic molecularly imprinted polymer nanocomposites (Fe_3_O_4_-MIP) modified on the surface of screen-printed carbon electrodes (SPCE)	20 nM	0.05–10 μM	[73]
	Combining magnetic nanomaterials Fe_3_O_4_NPs and HCR for simultaneous signal-guided electrochemical detection of miRNAs	miR-141:0.28 fM miR-21:0.36 fM	1 fM–1 nM	[74]
Metal-organic framework materials	Sensitivity detection of three isomers of hydroquinone, catechol, and resorcinol based on M@Pt@M-RGO electrochemical biosensor	HQ:0.015 μmol/L CT:0.032 μmol/L RS:0.133 μmol/L	HQ:0.05–200 μmol/L CT:0.1–160 μmol/L RS:0.4–300 μmol/L	[75]
	An electrochemical biosensor to detect simultaneously PA and DA using HKUST-1 (Cu-BTC) coupled with graphene oxide (ERGO)	PA:0.2–160 μM DA:0.2–300 μM	PA:0.016 μM DA:0.013 μM	[76]
	An electrochemical biosensor to detect UA using CeO2-x/C/RGO nanocomposites synthesized by MOF and graphene oxide	2.0 μmol/L	49.8–1050.0 μmol/L	[77]

**Table 2 biosensors-12-00566-t002:** Signal amplification strategies based on other materials.

Strategies	Examples	Limit of Detections	Linearity Ranges	Ref.
Enzyme	Electrochemical immunosensor based on DT-diaphorase (DT-D) as oxidoreductase labeling and 4-nitroso-1-naphthol (4-NO-1-N) as reaction substrate	PTH:2 pg/mL	2 pg/mL–1 μg/mL	[125]
	Electrochemical biosensor based on the display of tyrosinase on the surface of Escherichia coli cells for the detection of Bisphenol A	0.01 nm	BPA:0.01 nm–100 nm	[126]
Nucleic acid amplification	An electrochemical biosensor based on cyclic enzyme signal amplification (CESA) with DSN and 3-QD-DNA nanocomposites as cascade signal probes for hypersensitive detection of microRNA	1.2 amol/L	5 amol/L–5 fmol/L	[136]
	An electrochemical biosensor using double-stranded specific nuclease (DSN) and cleavage endonuclease (NEase) catalyzed reactions to detect miRNA	3 aM	10 aM–10 fM	[137]
	Ultra-sensitive detection of microRNA by an electrochemical biosensor based on RCA-mediated palladium nanoparticles (PdNPs)	8.6 amol/L	50 amol/L–100 fmol/L	[138]
	Protein detection by electrochemical biosensors based on molecular recognition between aptamer and target	0.17 pM	0.5 pM–300 nM	[143]
	Efficient detection of exosomal microRNAs by strand displacement reaction (SDR) based electrochemical biosensor	0.4 fM	miRNA-21:1 fM–200 pM	[144]
Polymers	Electrochemical biosensor based on methylene blue (MB) containing MnO_2_-functionalized COF, and metallic gold-platinum nanoparticles (AuPbNPs) for ultra-sensitive detection of PSA	16.7 fg mL^−1^	0.00005–10 ng mL^−1^	[153]
	Electrochemiluminescent immunosensor based on AMGMs nanocomposites for the detection of PSA in serum	0.1 pg/mL	PSA:0.1 pg/mL–50 ng/mL	[158]
Redox markers	An electrochemical biosensor to detect microRNA-21 using toluidine blue (TB) electrostatic adsorption aggregation signal amplification	78 amol/L	100 amol/L–1 nmol/L	[161]
	An electrochemical biosensor based on RuHex and screen-printed gold electrodes (SPGEs) to detect microRNA	100 amol/L	100 amol/L–100 pmol/L	[162]
Cell or tissue	The RSIT sensor by using rat small intestine tissue cells as a sensitive element and effector to detect resveratrol	1 × 10^−13^ mol/L	-	[172]
	Cell membrane biosensor with hTRPV1 immobilized directly on the HEK293T cell membrane to detect spicy substances	-	-	[188]

## Data Availability

This article has not been submitted to other journals, and the cited materials are labeled references.

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
