# Peer review of "Electrochemical Signal Amplification Strategies and Their Use in Olfactory and Taste Evaluation"

_biosensors, 2022, doi:10.3390/bios12080566_

Round 1

Reviewer 1 Report

Comments in the attached file

Reviewer 2 Report

The authors summarized electrochemical signal amplification strategies and research progress on their use in olfactory and taste determination. Although being interesting and informative, I find that there are some major issues with the paper that require addressing prior to this being considered for publication in this journal. I have identified the main points for consideration below:

1.This manuscript has some spelling typos, style errors and grammatical errors, which severely affect its readability. Pleases carefully check and correct them in the revised manuscript.

2.The challenge and future directions of electrochemical biosensors for olfactory and taste determination should be discussed in detail.

3. For a review manuscript, the detailed introduction of the relevant research progress is necessary, while the author's own views and objective evaluation are more important which are ignored, unfortunately.

4. The advantages of electrochemical biosensors should be introduced and some recent work should be cited, for example, Microchemical Journal 179 (2022) 107515, Journal of Hazardous Materials 436 (2022) 129107.

Round 2

Reviewer 1 Report

Accept

Reviewer 2 Report

The authors have addressed the comments. There is no further comment.